# Development of Root Caries Prevention by Nano-Hydroxyapatite Coating and Improvement of Dentin Acid Resistance

**DOI:** 10.3390/ma15228263

**Published:** 2022-11-21

**Authors:** Miyu Iwasaki, Ryouichi Satou, Naoki Sugihara

**Affiliations:** Department of Epidemiology and Public Health, Tokyo Dental College, Tokyo 101-0061, Japan

**Keywords:** nano-hydroxyapatite, fluoride, dentin, root caries, demineralization, microradiography

## Abstract

There is no established method for optimizing the use of dentin to prevent root caries, which are increasing in the elderly population. This study aimed to develop a new approach for root caries prevention by focusing on bioapatite (BioHap), a new biomaterial, combined with fluoride. Bovine dentin was used as a sample, and an acid challenge was performed in three groups: no fluoride (control group), acidulated phosphate fluoride treatment (APF group), and BioHap + APF treatment (BioHap group). After applying the new compound, the acid resistance of dentin was compared with that of APF alone. The BioHap group had fewer defects and an increased surface hardness than the APF group. The BioHap group had the smallest lesion depth and least mineral loss among all groups. Using a scanning electron microscope in the BioHap group showed the closure of dentinal tubules and a coating on the surface. The BioHap group maintained a coating and had higher acid resistance than the APF group. The coating prevents acid penetration, and the small particle size of BioHap and its excellent reactivity with fluoride are thought to have contributed to the improvement of acid resistance in dentin. Topical fluoride application using BioHap protects against root caries.

## 1. Introduction

Gingival recession is caused by periodontal disease, decreased salivary secretion, and inappropriate brushing, all of which contribute to increased root caries in the elderly [1,2,3]. Root dentin exposed to the gingival recession has low acid resistance and a high critical pH of 6.0–6.2, which has been reported to accelerate the progression of root caries [4]. Often, root caries have no subjective symptoms, such as spontaneous pain or cold-water pain, which allow caries to spread laterally and surround the entire tooth root [5,6,7,8]. This disease progression pattern makes treatment difficult and increases the risk of tooth loss. Loss of occlusion may lead to malnutrition, oral frailty, or reduced quality of life [9,10]. Dentin is difficult to remineralize because of its low mineral content compared with enamel [11]. Therefore, the prevention of root caries is important.

Fluoride applications, such as topical fluoride application with acidulated phosphate fluoride (APF), stannous fluoride, and fluoride mouth-rinsing with sodium fluoride have been reported to be effective in preventing root caries [1,12,13]. The preventive effect of fluoride application is lower for enamel caries because of the lower mineral content of the root surface dentin and the smaller amounts of calcium fluoride produced in the superficial layers of dentin. For this reason, fluoride application has a limited caries preventive effect [14].

In recent years, calcium phosphates, which are stores of calcium and phosphorus, have been studied to prevent root caries and a new prophylaxis strategy using this material in combination with fluoride has been developed [11,14,15,16]. When calcium phosphates are applied to the tooth surface, the calcium and phosphate ion concentrations in saliva maintain a supersaturated state, generating octacalcium phosphate, a precursor of hydroxyapatite (HAp) [17,18]. The HAp precursors are converted to HAp through repeated dissolution and reprecipitation, allowing mineral recovery [11,14,19]. A protective film of apatite coating on the tooth surface is reported to protect the tooth surface from acid and suppress demineralization [20,21]. The particle size of calcium phosphate produced by the reaction of fluoride and HAp is closely related to its caries preventive effect [22,23,24]. Calcium phosphate with a particle size of about 50 nm was reported to have a high reactivity with tooth structures and improve the demineralization suppression effect [23]. It has been shown that the smaller the particle size of calcium phosphate, the better the adsorption of minerals and fluoride, and the higher the caries prevention effect [22,23,24]. It has also been reported that nanoparticles smaller than 3–6 μm, which were the diameter of the dentinal tubules, could close dentinal tubules and were therefore beneficial in preventing dentin hypersensitivity [25].

This study focused on a new calcium phosphate formulation, bioapatite (BioHap; BIOAPATITE Inc., Shiga, Japan). A type BioHap is an eggshell-derived bioceramic with a particle size of less than 50 nm and a structural formula of (Ca:Mg)_10_(PO_4_)_6_(OH)_2_ (Japanese Laid-Open Patent Publication No.2020-105060 and 2020-117423). This compound has a similar chemical composition to HAp and is bioactive, has osteoinductivity, and is resistant to ultraviolet and X-rays [26,27,28,29]. It is characterized by the presence of magnesium (Mg), which is not present in the calcium phosphates currently used clinically. Mg is abundant in hard tissues, such as bones, teeth, and living bones. Mg activates osteoblasts and osteoclasts and promotes bone cell metabolism, thereby enhancing biocompatibility [30]. In addition, Kodaka et al. and Schroeder et al. reported that Mg is involved in forming whitlockite, which is required to convert HAp [31,32].

In this study, we focused on the effects of applying calcium phosphate to root surface dentin, protection of the tooth surface, the conversion of HAp to fluorapatite, and the formation of whitlockite from Mg. This study aimed to develop a new root caries prevention method by focusing on bioapatite (BioHap), a new biomaterial, and combining BioHap with a fluoride coating. In addition, using techniques such as surface texture measurement, scanning electron microscope (SEM), and contact microradiography (CMR), the acid resistance of dentin after the application of the new preventive method was evaluated in comparison with the conventional topical fluoride application.

## 2. Materials and Methods

### 2.1. Preparation of Bovine Tooth Dentin Block Samples

Our sample consisted of the cervical third of the root dentin of 18 bovine mandibular anterior teeth. The dentin blocks were prepared using water-resistant abrasive paper (#600) to form a smooth outline of labial dentin, which was then mirror-polished using water-resistant abrasive paper (#1000, #2000, and #4000). A window of 5 mm width × 5 mm depth × 10 mm height was formed using inlay wax on the labial side of the dentin block.

### 2.2. Acid Challenge Experiment

The samples were divided into three groups: no fluoride (control group), APF treatment only (APF group), and BioHap + APF treatment (BioHap group). Nine samples from each group were used for the experiment (*n* = 9). In the BioHap group, after applying BioHap, 10% phosphoric acid gel (pH 1.0) was added for 1 min, and the blocks were immersed in a phosphate acid-sodium fluoride solution (APF, 9000 ppmF, pH 3.6) for 4 min. For the APF group, a conventional 4-min topical fluoride application was used. There was no fluoride application in the control group. After preventive treatment, the measurement samples were immersed in a remineralization solution (0.02 M HEPES-based buffer solution, Ca: 3 mM, P: 1.8 mM, pH 7.3, DS: 10) for 1 h at 37 °C and a demineralization solution (0.1 M Lactic acid-based buffer gel, Ca: 3 mM, P: 1.8 mM, pH 5.0, DS: 10) for 1 h at 37 °C.

### 2.3. 3D Measurement Laser Microscopic Observation

After the acid challenge, each sample was immersed in xylene and dehydrated using ethanol. The samples were analyzed using a 3D laser microscope (LEXT OLS4000, Olympus Co., Ltd., Tokyo, Japan). We measured the difference in the number of substantial defects between the control surface that had not been demineralized by wax-up and the experimental surface that had been demineralized. The samples were also measured to calculate the experimental and control surfaces’ average roughness (Sa). The measurement area was 645 × 645 µm, and the cutoff value was 80 µm. The number of substantial defects and Sa were measured at five points per sample on the boundary between the control and experimental surfaces, and the mean ± standard deviation (SD) was calculated.

### 2.4. Micro Vickers Hardness Test

Micro Vickers hardness measurements were performed using a Micro Vickers hardness tester (HMV-1, Shimadzu Co., Ltd., Tokyo, Japan). The tester was set to a pressing load of 0.49 N for 20 s on the control and experimental samples. The Micro Vickers hardness was measured at three points per sample, and the average value ± standard deviation (SD) was calculated.

### 2.5. Scanning Electron Microscope Observation

The samples were subjected to carbon deposition during pre-measurement preparation using a vacuum deposition device (VC-100S; Vacuum Device Co., Ltd., Ibaraki, Japan). The samples were embedded in polyester resin (Rigolac; Nisshin EM Co., Ltd., Tokyo, Japan), mirror-polished, and observed in cross sections. The surfaces of their samples were observed using an SEM (SU6600; Hitachi Co., Ltd., Tokyo, Japan) at a voltage of 15 kV. The photographing magnification was 10,000× for surface observation and 5000× for cross-sectional observation.

### 2.6. Contact Microradiography

The samples were cut into polished sections with a thickness of 100 μm. Using a soft X-ray generator (CMR-3; Softex Co., Ltd., Tokyo, Japan) equipped with a 20 μm thick Ni filter, a 20 μm step aluminum step wedge was set to distinguish from step 1 to step 20. The settings were a tube voltage of 15 kV, tube current of 3 mA, and irradiation time of 6–9 min. Observations were performed using an optical microscope at 200× magnification. A high-precision glass plate (HRP-SN-2; Konica Minolta Inc., Tokyo, Japan) was used for photography.

The glass plate was developed in a developer (D-19; Kodak, Rochester, NY, USA) at 20 °C for 5 min and fixed (Super Fujifix-L; Fujifilm Co., Ltd., Tokyo, Japan) at 20 °C for 5 min. The glass plate was washed with water for 10 min and dried. The completed glass plate was converted to grayscale (8 bit, 256 gradations) using an image analysis software (Image Pro Plus, version 6.2; Media Cybernetics Inc., Silver Spring, MD, USA) and an image analysis system (HC-2500/OL; Olympus Co., Ltd., Tokyo, Japan) to acquire the density profile.

The lesion depth (Ld) and mineral loss value (ΔZ) were measured at five sites in a range of 50 × 200 μm from the tooth’s surface to the deep part of the healthy dentin. The distance from the dentin surface to the part where the mineral content in the lesion area accounts for 95% of the mineral content in healthy dentin [33]. ΔZ was based on the aluminum step wedge taken simultaneously with the density of the sample, and the mineral loss was calculated using the formula of Angmar et al. [33,34]. The values were converted into a histogram with mineral values of 0% and healthy dentin as 100% [33].

### 2.7. Statistical Analysis

The number of substantial defects, Sa, and Micro Vickers hardness for the three groups of preventive treatment were measured as the mean ± SD of nine samples. Ld and ΔZ were measured as the mean ± SD of five samples, excluding damaged samples during the operation. Comparison of preventive treatment among the three groups was calculated by one-way analysis of variance (ANOVA), with *p* values determined as significant at *p* < 0.05. The Bonferroni test was used for post hoc comparisons when an ANOVA was significant (*p* < 0.05, 0.001, respectively). Graph creation and data analysis were performed using software (ORIGIN 2022b, Lightstone Co., Ltd., Tokyo, Japan).

## 3. Results

### 3.1. Amount of Substantial Defect after an Acid Challenge by the 3D Measurement Laser Microscope

Figure 1 shows 3D laser microscope images of substantial defects in dentin after an acid challenge. Figure 1a–c shows the dentin surface’s non-demineralized control plane on the image’s left side and the demineralized experimental plane on the right side. Figure 1d shows the mean and SD for each group. The control group had a more significant deficit due to demineralization, 1.821 ± 0.025 μm when compared with the control group (*p* < 0.001) (Figure 1a,d). The APF group showed a parenchymal defect of 0.921 ± 0.024 μm, but the amount of substantial defect was reduced compared to that in the control group (*p* < 0.001) (Figure 1b,d). The BioHap group had the smallest substantial defect reduction (0.452 ± 0.042 μm) among all groups (*p* < 0.001) (Figure 1c,d).

### 3.2. Calculated Average Roughness after an Acid Challenge by the 3D Measurement Laser Microscope and Micro Vickers Hardness Measurements by Micro Vickers Hardness Tester

Figure 2 shows the results of the mean ± SD of Sa and Micro Vickers hardness on the experimental surface of dentin after the acid challenge. Sa was the smallest in the control group, with a value of 0.128 ± 0.012 μm. The diameter in the APF group was 0.203 ± 0.016 μm. The Sa of the BioHap group was 0.933 ± 0.286 μm, and an increase in Sa was observed. No significant difference was observed between the control and APF groups (*p* > 0.05) (Figure 2a). A significant difference was observed between the control and BioHap groups and between the APF and BioHap groups (*p* < 0.05 for both) (Figure 2a). The Micro Vickers hardness was the smallest in the control group at 35.487 ± 2.956 HV. The APF group had 42.349 ± 3.845 HV, which was significantly greater than that of the control group (*p* < 0.001) (Figure 2b). The BioHap group had 51.697 ± 2.855 HV, which was the largest value among all the groups, and a significant difference was observed between the groups (*p* < 0.001) (Figure 2b).

### 3.3. Dentin Surface and Cross-Section Scanning Electron Microscope Observations after Acid Challenge

Figure 3 shows the secondary electron images of the dentin surface (×10,000) and sagittal section (×5000) after the acid challenge. A surface SEM image of the control group showed enlargement of the dentinal tubule openings (Figure 3a). In the APF group, a narrowing and partial closure of the dentinal tubule opening, and the deposition of spherical products on the surface were observed (Figure 3b). The BioHap group showed closure of the dentinal tubule openings due to the deposition of more abundant and larger spherical and amorphous products on the dentin surface than in the APF group (Figure 3c).

Cross-section SEM images of the control group showed demineralization of the dentin surface layer and expansion of dentinal tubules (Figure 3d). In the APF group, a thin acid-resistant layer with a thickness of approximately 1–3 μm was formed on the dentin’s outermost layer. In this group, a demineralized image with a surface layer maintained on the inner tubule wall of the dentinal tubules was observed. However, the tubules were slightly expanded (Figure 3e). The BioHap group developed a 3–5 μm thick coating layer on the dentin surface. Spherical particles were observed on the surface of the coatings (Figure 3f). The inside of the coating was not uniform, and a thin demineralized layer and an acid-resistant layer were observed between the coating and dentin surface layer. The dentinal tubules showed less tubule expansion than those in the APF group. Calcified deposits were observed inside tubules (Figure 3f).

### 3.4. Measurement of Lesion Depth and Mineral Loss Value by Contact Microradiography Analysis

Figure 4 shows a CMR image of the experimental dentin surface after the acid challenge. In the control group, substantial defects due to demineralization were observed horizontally from the dentin surface layer, and gray to black demineralization images were observed 5–15 μm from the surface layer (Figure 4a). The dentin surface layer was maintained in the APF group. Some expansion of dentinal tubules was observed in the range of 5–10 μm from the surface, but demineralization was mild (Figure 4b). In the BioHap group, a thick coating-like layer with the same strength as that of healthy dentin was present on the dentin surface layer, and the thickness of the layer was locally different and uneven inside (Figure 4c). No demineralized image was observed in the dentin directly under the coating. The signal intensity was comparable to healthy dentin at a depth of ≥200 μm from the surface layer (Figure 4c).

Figure 5 shows a graph of the mineral profile at a depth of 20–25 µm for each group after the acid challenge. A depth of 150–200 μm shows the relative mineral value at each depth when the average value is 100%. In the control group, the increase from 25 μm was gradual, indicating that the minerals were lost over a wide range (Figure 5). In the APF group, the rising angle sharply improved, and the mineral value of the APF group was high at any depth in the control group. In the BioHap group, the increasing rise and apeak of over 100% were observed near 30–35 μm, maintaining a high mineral value of 95% or more (Figure 5).

Figure 6 shows graphs of each group’s average values and standard deviations of lesion depth (Ld) and mineral loss value (ΔZ) calculated by CMR analysis. The Ld of the control group was the largest at 65.395 ± 7.972 μm, and the APF group showed a decreasing trend to 57.023 ± 7.247 μm, but the differences were not significant (*p* > 0.05) (Figure 6a). The BioHap group was the smallest of all groups at 40.465 ± 2.867 μm, and this was significantly different than both the significant difference between the control and APF group (*p* < 0.05) (Figure 6a).

The ΔZ of the control group was 5635.143 ± 387.001 vol% × µm. The APF group showed a significant decrease to 5075.751 ± 298.689 vol% × µm (*p* < 0.05) (Figure 6b). In contrast, the BioHap group measured 4723.856 ± 238.578 vol% × μm, the smallest among the groups. There was no significant difference between the APF and BioHap groups, but there was a significant difference between the control group and the BioHap group (*p* < 0.05) (Figure 6b).

## 4. Discussion

In the SEM image of the dentin surface, the APF group showed deposition of spherical products (Figure 3b), and the BioHap group showed deposition of more large-diameter spherical and amorphous products than the APF group (Figure 3c). Ogaard et al. and Petzold et al. reported that when a high concentration of fluoride (1000 ppmF) or higher reacts with the tooth structure, calcium fluoride (CaF_2_)-like particles with a spherical structure are formed on the tooth surface [35,36]. Huang et al. reported that a transmission electron microscope observation image after applying nano-hydroxyapatite to the tooth surface showed a cylindrical shape with a diameter of 10–20 nm and a length of 60–80 nm, extending along the *c*-axis [24]. The spherical products observed in the surface SEM images of the APF and BioHap groups in this study were presumed to be CaF_2_-like particles. The cylindrical and polygonal amorphous particles observed in the BioHap group were presumed to be calcium phosphate. In this study, the surface SEM image of the BioHap group revealed many irregularly shaped particles of different sizes instead of uniform spherical particles, such as APF (Figure 3b,c). Ga et al. showed that the difference in the density of the calcium-phosphorus complex causes a change in the particle size of the CaF_2_ produced [37]. When the phosphate gel and BioHap were mixed in the BioHap group, sites with different densities of calcium and phosphate ions were scattered. One explanation is that the diameter of the CaF_2_-like particles produced changed when reacting with high-concentration fluoride ions in the next step.

In the BioHap group, a 3–5 μm thick coating layer was observed on the dentin surface (Figure 3f). Saxegaard et al. reported that the solubility of apatite increases as the particle size decreases and that the amount of CaF_2_ in the oral cavity affects acid resistance [24,38,39]. A coating of about 1 μm has been observed in a preventive method that uses calcium phosphate with a particle size of several tens of μm and fluoride in combination [40]. It is believed that the thickness of the coating depends on the particle size. BioHap has an extremely small particle size of ≤50 nm and is highly reactive. Therefore, it is likely that a large number of calcium phosphate-like particles and CaF_2_-like particles were generated, and thick coating-like observations appeared (Figure 3e). The thickness of the coating produced in the BioHap group was 2–3 μm thicker than that in the APF group, suggesting that the BioHap group had a higher acid resistance than the APF group. CaF_2_ and calcium phosphate produced after fluoride application to the human oral cavity have been reported to act as reservoirs for calcium, phosphorus, and fluoride [11,41,42]. The BioHap coating was produced in the presence of abundant calcium ions, phosphate, and fluoride ions from the composition of the demineralization/remineralization solution in this experimental system. It can be predicted that the above-mentioned ions are abundant in the coating and maintain a supersaturated state in the oral cavity by self-disintegration against acid stimulation. Consequently, it is thought that this contributes to the suppression of demineralization.

The Micro Vickers hardness after the acid challenge was the highest in the BioHap group, and ΔZ was the lowest in all groups (Figure 2a and Figure 6b). Huang et al. reported that calcium phosphate-like and CaF_2_-like crystals in the surface layer of dentin promote the formation of HAp, hydrofluoric apatite, fluorapatite, etc., and recover the surface hardness and mineral value of dentin [11,15,17,24]. In addition, Leal et al. clarified that casein phosphuretted-amorphous calcium phosphate and nano-hydroxyapatite applied to the tooth surface form a semipermeable membrane on the tooth surface and exert an inhibitory effect on demineralization [16,20].

In this study, when the dentin and BioHap group products were dissolved, a large amount of calcium, phosphate, and fluoride ions were diffused into the surroundings. In the dentin just below the coating, the saturation of various ions in the surface layer of the dentin increased, the conversion of HAp, hydrofluoric apatite, and fluorapatite was promoted due to chemical equilibrium, and the dentin hardness and mineral value were recovered. In addition, until the coating of the BioHap group was completely dissolved and the tooth was exposed to acid, protected the acid from penetrating the dentin surface layer and crystals and suppressed demineralization. The CMR results showed no significant difference in ΔZ between the APF and BioHap groups (Figure 6b), which may result from the composition of BioHap, which contains a large amount of Mg. Tung et al. and Saito et al. reported that dentin contains a large amount of Mg and carbonic acid, so easily soluble whitlockite is likely to form [43,44]. The abundant Mg content in dentin and the BioHap group promoted the formation of whitlockite and made it intolerable to demineralization. Therefore, it was presumed that no significant difference was observed between the APF and BioHap groups.

The increase in Sa in the BioHap group may be due to the generated calcium phosphate-like particles and CaF_2_-like particles (Figure 2a). Cury et al. and Boollen et al. showed that when Sa exceeds 0.2 μm, the risk of caries and periodontal disease increases [45,46]. The Sa in the BioHap group was 0.933 ± 0.286 µm, exceeding 0.2 µm. Therefore, plaque accumulation may increase after applying this new preventive method. Consequently, it is necessary to develop clinical treatments to reduce the roughness.

This study is the first in vitro experiment in which BioHap was used with fluoride to develop preventive methods. In this experiment, the composition of the artificial saliva was as simple as possible, and salivary proteins and bacteria were excluded. For future clinical applications, collecting in situ experimental data that consider the effects of saliva and biofilms is desirable. Calcium phosphate can deteriorate due to various factors [28,29,47]. The effects of aging on coatings containing calcium phosphate must also be considered. In addition, observation of the crystal structure by the transmission electron microscope and analysis by X-ray diffraction is necessary to identify the substance that constitutes the coating. To predict the prognosis after treatment, we evaluated the physical strength of the coating and its resistance to brushing stimuli.

## 5. Conclusions

Thick acid-resistant coatings consisting of calcium phosphate-like and CaF_2_-like particles were observed in dentin after the new prophylaxis developed in this study. The BioHap group suggested a significant improvement in acid resistance compared to conventional methods of prophylaxis, such as a decrease in the amount of substantial defect, an increase in Micro Vickers hardness, and a decrease in Ld and ΔZ due to CMR. We have demonstrated that our developed prophylaxis is a strong candidate for the prevention of root caries. This novel method, which combines BioHap, a new biomaterial, with fluoride tooth surface application, is expected to be clinically applied as a new preventive method for root caries.

## Figures and Tables

**Figure 1 materials-15-08263-f001:**
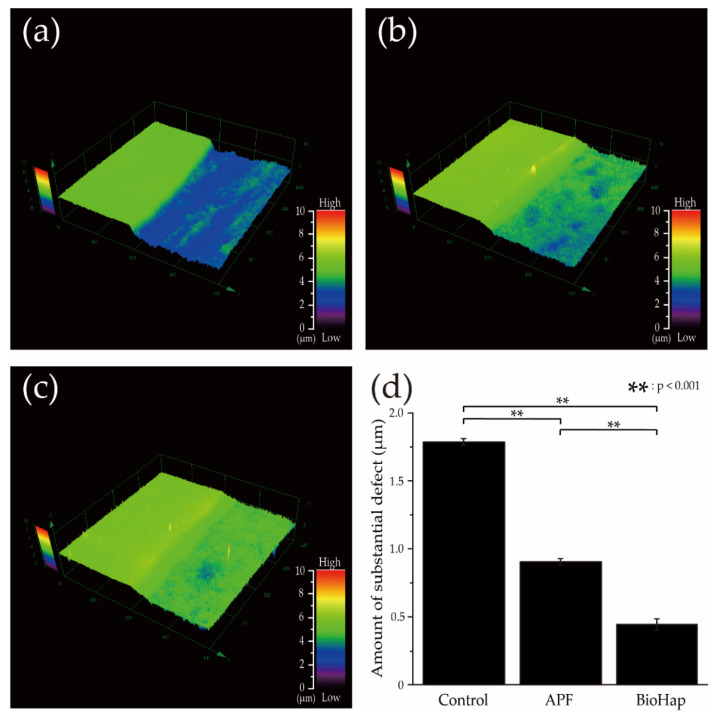
Comparison of non-decalcified and decalcified interface images and measurements of substantial defects after the acid challenge. (**a**) Control, (**b**) APF, and (**c**) BioHap groups. (**a**–**c**) show the non-decalcified control surface on the left and decalcified experimental surface on the right. The contrast plane was used as the reference plane. (**d**) Average ± SD of substantial defect (*n* = 9). The horizontal axis indicates the various preventive treatments, and the vertical axis indicates the step (μm) between the control and experimental surfaces. Color scale bars represent 0–10 μm.

**Figure 2 materials-15-08263-f002:**
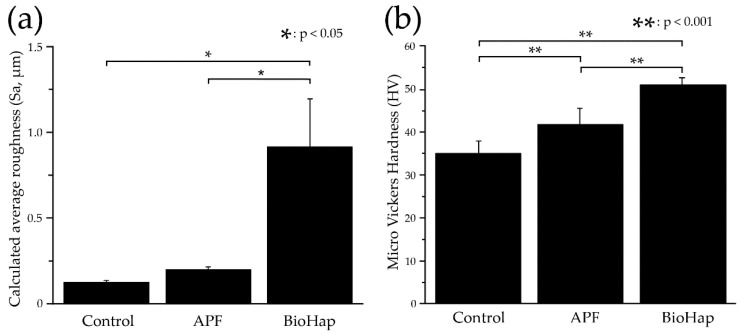
Comparison of calculated average roughness (Sa) and Micro Vickers hardness measurements of experimental surfaces after acid challenge. (**a**) Graph of Sa (*n* = 9). The horizontal axis indicates various preventive treatments, and the vertical axis indicates the Sa (μm) of the experimental surface. (**b**) Graph of Micro Vickers hardness (*n* = 9).

**Figure 3 materials-15-08263-f003:**
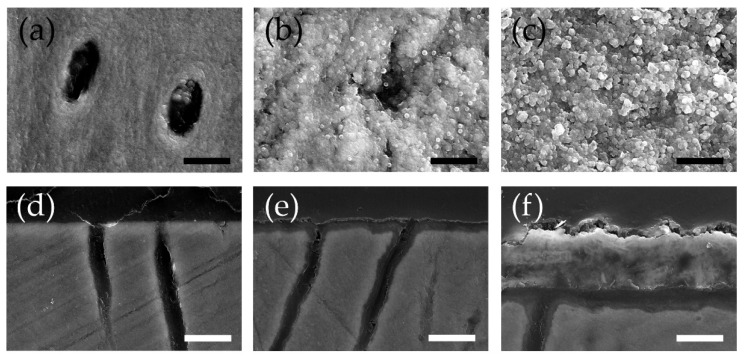
Images of surface and cross-section scanning electron microscope (SEM) observations after acid challenge. The top images show surface SEM observations taken at 10,000 magnifications. From the left, (**a**) control group, (**b**) APF group, and (**c**) BioHap group. The black scale bar represents 2.5 μm. The lower images show cross-section SEM observations taken at 5000 magnifications. From the left, (**d**) control group, (**e**) APF group, and (**f**) BioHap group. The white scale bar represents 5.0 μm.

**Figure 4 materials-15-08263-f004:**
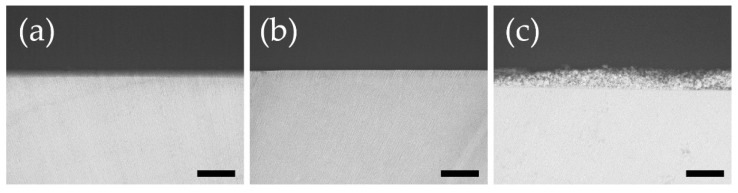
Optical microscopy image of contact microradiography taken at 200 magnifications after acid challenge. From the left, control group (**a**), APF group (**b**), and BioHap group (**c**). Black scale bar represented 100 μm.

**Figure 5 materials-15-08263-f005:**
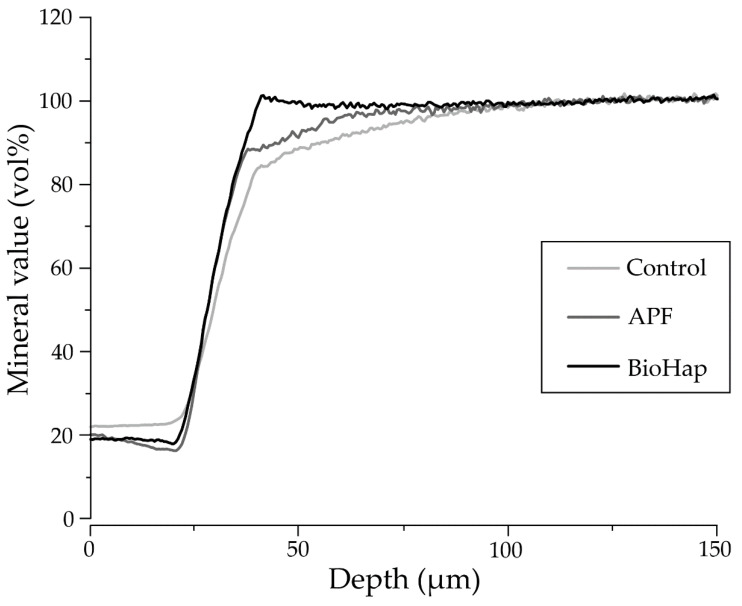
Graph of the mineral profile of preventive treatments for each group after the acid challenge. The horizontal axis indicates the lesion depth, and the vertical axis shows the mineral value (vol%). As for the settings, the lesion depth demineralization start line is set to 25 μm, and the mineral value start line is set to 20–25 vol%.

**Figure 6 materials-15-08263-f006:**
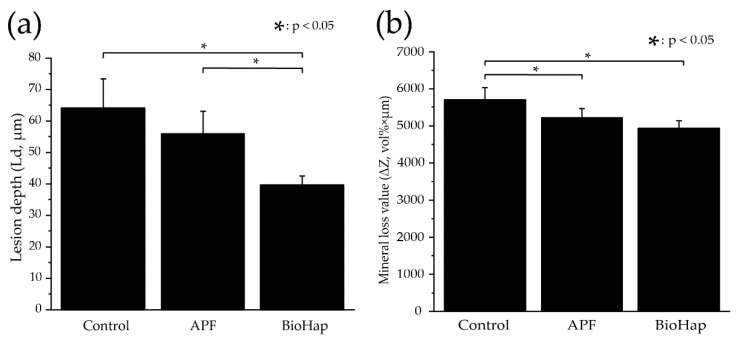
Comparison of measured lesion depth (Ld) and mineral loss value (ΔZ) by contact microradiography after acid challenge. (**a**) Shows the different test groups for Ld (*n* = 5). The horizontal axis indicates various preventive treatments, and the vertical axis indicates the Ld (μm) of the experimental surface. (**b**) Shows the different test groups for ΔZ (*n* = 5). The horizontal axis indicates various preventive treatments, and the vertical axis indicates the ΔZ (μm) of the experimental surface.

## Data Availability

All data are included in the article.

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
