# Peer review of "Development of Root Caries Prevention by Nano-Hydroxyapatite Coating and Improvement of Dentin Acid Resistance"

_materials, 2022, doi:10.3390/ma15228263_

Round 1
Reviewer 1 Report
Dear authors
This is an interesting work proposing a new approach using bioapatite combined with fluoride for increasing the acid resistance of dentin, aiming to create an efficient treatment for root caries prevention. The results presented herein were able to show that the bioapatite-fluoride group produced the lower dentin mineral loss and produced the obliteration of dentinal tubules, which are evidences of dentin acid resistance increasing. These aspects represent a novelty and encourage the idea for the article publication.
Author Response
> Thank you very much for your excellent suggestion.
We would like to express our strong gratitude for your insightful comments on our paper. With this comment, we feel that we would like to push forward with further research. The paper has been edited and rewritten by an experienced scientific editor, who has improved the grammar and stylistic expression of the paper.
Reviewer 2 Report
Dear authors,
I read your manuscript and I found that your design experimental is so simple. I suggested you to try search on internet and you can find that there are many in vitro tests which can help you to improved your manuscript to confirm your achievements. Any way, your manuscript is so simple and your findings needs more test for being confirmed.
Author Response
> We appreciate the reviewer's concerns on this point. Observation of the coating is the key point in this study. In the method of combining calcium phosphate formulation and fluoride application used in many previous studies, almost no coating was observed even though it suggested suppression of demineralization. The new preventive method we developed has a thick coating on the dentin surface layer observed in cross-section SEM and CMR images. Since the bioapatite used in this study has low crystallinity and fine particles, it has enhanced adsorption power and can be easily incorporated into other substances. It also contains magnesium, so it is safe to use without any harmful effects. We speculate that bioapatite's properties increased its reactivity with fluoride, forming a thick coating on the dentin surface layer. Due to the presence of this coating, the new preventive method is considered to have closed dentinal tubules and improved acid resistance.
We also agree that additional information such as that suggested by the reviewers would be valuable in confirming our findings. Regrettably, however, because of we do not have experimental technology, we are unable to do the experimentation. In the future, we plan to acquire techniques such as the transmission electron microscope and X-ray diffraction, and to observe the crystal structure of the coating and elucidate changes in the microstructure of dentin.
Reviewer 3 Report
Refereree report on manuscript “Development of root caries prevention by nano-hydroxyapatite coating and improvement of dentin acid resistance”
This version does not look worthy and cannot be recommended for publication in this form and at least needs major revision.
1. The introduction mainly discusses information from Dental journals, while this “Materials” journal requires more information about material properties. Because of this, the motivation and relevance of this study may be in doubt, because the latest achievements in HAP material research and development are not reflected. This information should be updated, for example, using the search for the latest achievements: https://www.mdpi.com/search?q=hydroxyapatite
2. For dental applications , it is necessary to note the resistance of the HAP to both ultraviolet and X- irradiations. The outcome of HAP indeed depends on their resistance to aging, including radiation. See:
Bystrova, A.; Dekhtyar, Y.D.; Popov, A.; Coutinho, J.; Bystrov, V. Modified hydroxyapatite structure and properties: Modeling and synchrotron data analysis of modified hydroxyapatite structure. Ferroelectrics 2015, 475, 135–147.
Hübner, W.; Blume, A.; Pushnjakova, R.; Dekhtyar, Y.; Hein, H.-J. The influence of X-ray radiation on the mineral/organic matrix interaction of bone tissue: An FT-IR microscopic investigation. Int. J. Artif. Organs 2005, 28, 66–73.
3. Line 122-128. What was the dynamic range of radiographic tests? What is the average radiation dose?
4. Lines 267-268. “3.3. Formatting of Mathematical Components 267 This is example 1 of an equation: “ There is a feeling that this paragraph is not finished and that something is not finished there.
5. In the conclusions, it is necessary to clearly formulate what new data about the studied materials were obtained in this work? This is somehow connected with the relevance and novelty of research, which, unfortunately, is not disclosed perfectly in the introduction.
In general, the manuscript is interesting and can be considered for publication after constructive reflection on the above comments.
Author Response
1.The introduction mainly discusses information from Dental journals, while this “Materials” journal requires more information about material properties. Because of this, the motivation and relevance of this study may be in doubt, because the latest achievements in HAP material research and development are not reflected. This information should be updated, for example, using the search for the latest achievements: https://www.mdpi.com/search?q=hydroxyapatite
> Thank you very much for your valuable opinion. We agreed with the reviewers' opinions and added the latest research and development of HAP materials to the paper.
Page 2, Line50-54
Clinically used nano-hydroxyapatite is a highly bioactive and biocompatible calcium phosphate with the same morphology and crystal structure as HAp in the dental hard tissue [22]. In addition, because the particle size is nano-sized, it is reported that it has excellent reactivity with fluoride and is highly effective in preventing caries [23,24].
[22] Amaechi, B.T.; Alshareif, D.O.; Azees P.A.A.; Shehata, M.A.; Lima, P.P.; Abdollahi, A.; Kalkhorani, P.S.; Evans, V.; Bagheri, A.; Okoye, L.O. Anti-caries evaluation of a nano-hydroxyapatite dental lotion for use after toothbrushing: An in situ study. J Dent. 2021, 115, 103863. DOI: 10.1016/j.jdent.2021.103863.
[23] Juntavee, A.; Juntavee, N.; Sinagpulo, A.N. Nano-Hydroxyapatite Gel and Its Effects on Remineralization of Artificial Carious Lesions. Int J Dent, 2021, 2021, 7256056. DOI: 10.1155/2021/7256056.
[24] Balasooriya, I.L.; Chen, J.; Korale Gedara, S.M.; Han, Y.; Wickramaratne, M.N. Applications of Nano Hydroxyapatite as Adsorbents: A Review. Nanomaterials, 2022, 12(14), 2324. DOI: 10.3390/nano12142324.
- For dental applications , it is necessary to note the resistance of the HAP to both ultraviolet and X- irradiations. The outcome of HAP indeed depends on their resistance to aging, including radiation.
> We agree that additional information on the resistance of the HAP to both ultraviolet and X- irradiations as the reviewer suggested would be valuable. Regrettably, however, because of we do not have equipment such as X-ray diffraction or experimental technology that can analyze changes in the microstructure of dentin, we are unable to do the experimentation.
- Line 122-128. What was the dynamic range of radiographic tests? What is the average radiation dose?
> The microradiography used in this experiment uses characteristic X-rays of only Kα rays with Cu as the anode material, and the radiation dose is very small. Furthermore, we use a Ni filter to cut Kβ rays. The irradiation area of CMR is about half that of film used in dentistry, and the radiation dose is less than 0.01 mSv.
> Contact microradiography (CMR) is a method with a wide dynamic range [1]. In this experiment, we used a glass film plate and a developing solution instead of the imaging plate (IP) used in medical X-ray examinations. Since it is an analog system, it is considered that saturation is unlikely to occur. As for the dynamic range, at least 20 steps can be discriminated with the reference system's aluminum step wedge.
[1] Elliott, J.C.; Wong, F.S.; Anderson, P.; Davis, G.R.; Dowker, S.E. Determination of mineral concentration in dental enamel from X-ray attenuation measurements. Connect Tissue Res. 1998, 38, 61-72.
- Lines 267-268. “3.3. Formatting of Mathematical Components 267 This is example 1 of an equation: “ There is a feeling that this paragraph is not finished and that something is not finished there.
> Thank you for providing these insights. This error has been corrected in accordance with the reviewer's comment. (Page 8, Line 270-271.)
5.In the conclusions, it is necessary to clearly formulate what new data about the studied materials were obtained in this work? This is somehow connected with the relevance and novelty of research, which, unfortunately, is not disclosed perfectly in the introduction.
> In accordance with the reviewer's comment, we have added this sentence to conclusion:
Page 9, Line347-352
The BioHap group suggested a significant improvement in acid resistance compared to conventional methods of prophylaxis, such as a decrease in amount of substantial defect, an increase in micro vickers hardness, and a decrease in Ld and ΔZ due to CMR.
Round 2
Reviewer 2 Report
Dear Editor,
I again check and review this manuscript, I am still in my previous comments, this manuscript need more tests to confirm their achievements.
My previous comments:
I read your manuscript and I found that your design experimental is so simple. I suggested you to try search on internet and you can find that there are many in vitro tests which can help you to improved your manuscript to confirm your achievements. Any way, your manuscript is so simple and your findings needs more test for being confirmed.
Author Response
We wish to thank the reviewer for this comment.
I again check and review this manuscript, I am still in my previous comments, this manuscript need more tests to confirm their achievements.
My previous comments:
I read your manuscript and I found that your design experimental is so simple. I suggested you to try search on internet and you can find that there are many in vitro tests which can help you to improved your manuscript to confirm your achievements. Any way, your manuscript is so simple and your findings needs more test for being confirmed.
> Thank you for your interest in additional information regarding the experimental design of this study. I agree with the reviewer's comments. However, the experimental design of this study is a gold standard method for determining tooth demineralization and remineralization. As a result of a literature search, three tests, profilometry, nanoindentation, and microradiography, are the most applied quantitative methods for judging acid resistance of teeth and scanning electron microscopy is the most major qualitative method. Many previous studies have combined one or two quantitative and qualitative analyses, and have not used as many analyzes as this study. In this study, three quantitative methods and one qualitative method are used to analyze acid resistance from various angles. Could you please provide specific experimental methods that you think should be added?
We also searched the Internet for similar previous studies, but we consider our study to be novel compared to those studies. As a result of quantitative and qualitative analysis in this study, it was clarified that the method developed by us has improved acid resistance compared to the conventional method. It is thought that the fine particles of BioHap were highly reactive with fluoride and formed a thick film. It is thought that the coating kept the dentin surface in a supersaturated state and physically inhibited the attack of acid.
We agreed with the reviewers' comments. We added sentence and reference which comparison with the previous studies.
Page 8, Line 293-295
A coating of about 1 μm has been observed in a preventive method that uses calcium phosphate with a particle size of several tens of μm and fluoride in combination [39]. It is believed that the thickness of the coating depends on the particle size.
Reference
[39] Miki, N.; Miake, Y.; Shimoda, S.; Mishima, H. Evaluation of Enamel Acid Resistance and Whitening Effect of the CAP System. Dent J (Basel). 2022, 10(9), 161. DOI: 10.3390/dj10090161.
Reviewer 3 Report
- For dental applications , it is necessary to note the resistance of the HAP to both ultraviolet and X- irradiations. The outcome of HAP indeed depends on their resistance to aging, including radiation.......
this comment, which is partly copied from the first report, is important and should be reflected in the introduction, even if the authors are currently unable to perform appropriate measurements
Author Response
- For dental applications , it is necessary to note the resistance of the HAP to both ultraviolet and X- irradiations. The outcome of HAP indeed depends on their resistance to aging, including radiation.......
this comment, which is partly copied from the first report, is important and should be reflected in the introduction, even if the authors are currently unable to perform appropriate measurements.
> We wish to express our deep appreciation to the reviewer for your insightful comment on this point. Regarding this comment, we agreed with the reviewer's opinion and added the following sentence to the “Introduction” and “Discussion” section. In addition, I have quoted the references regarding the resistance and aging of HAp that you told me round 1. Thank you very much for providing valuable references.
Page 2, Line 55-57
This compound has a similar chemical composition to HAp and is bioactive, osteoinductivity and resistant to ultraviolet and X-rays [24–27].
Page 9, Line 342-344
Calcium phosphate can deteriorate due to various factors [26,27,46]. The effects of aging on coatings containing calcium phosphate must also be considered.
Reference
[24] Alanis-Gómez, R.P.; Rivera-Muñoz, E.M.; Luna-Barcenas, G.; Alanis-Gómez, J.R.; Velázquez-Castillo R. Improving the Mechanical Resistance of Hydroxyapatite/Chitosan Composite Materials Made of Nanofibers with Crystalline Preferential Orientation. Materials (Basel). 2022, 15(13), 4718. DOI: 10.3390/ma15134718.
[25] Amaechi, B.T.; Alshareif, D.O.; Azees P.A.A.; Shehata, M.A.; Lima, P.P.; Abdollahi, A.; Kalkhorani, P.S.; Evans, V.; Bagheri, A.; Okoye, L.O. Anti-caries evaluation of a nano-hydroxyapatite dental lotion for use after toothbrushing: An in situ study. J Dent. 2021, 115, 103863. DOI: 10.1016/j.jdent.2021.103863.
[26] Bystrova, A.; Dekhtyar, Y.D.; Popov, A.; Coutinho, J.; Bystrov, V. Modified hydroxyapatite structure and properties: Modeling and synchrotron data analysis of modified hydroxyapatite structure. Ferroelectrics. 2015, 475, 135–147. DOI: 10.1080/00150193.2015.995580.
[27] Hübner, W.; Blume, A.; Pushnjakova, R.; Dekhtyar, Y.; Hein, H.-J. The influence of X-ray radiation on the mineral/organic matrix interaction of bone tissue: An FT-IR microscopic investigation. Int J Artif Organs. 2005, 28, 66–73. DOI: 10.1177/039139880502800111.
[46] Özdemir, O.; Kopac, T. Recent Progress on the Applications of Nanomaterials and Nano-Characterization Techniques in Endodontics: A Review. Materials (Basel). 2022, 15(15), 5190. DOI: 10.3390/ma15155109.
Round 3
Reviewer 3 Report
It can be seen that the authors have now quite improved their original manuscript, which now can be accepted for publication.
Author Response
It can be seen that the authors have now quite improved their original manuscript, which now can be accepted for publication.
> Thank you very much for providing important comments. We are thankful for the time and energy you expended.